# Fluorine-Nitrogen-Codoped Carbon Dots as Fluorescent Switch Probes for Selective Fe(III) and Ascorbic Acid Sensing in Living Cells

**DOI:** 10.3390/molecules27196158

**Published:** 2022-09-20

**Authors:** Shuai Ye, Mingming Zhang, Jiaqing Guo, Xiantong Yu, Jun Song, Pengju Zeng, Junle Qu, Yue Chen, Hao Li

**Affiliations:** Shenzhen Key Laboratory of Photonics and Biophotonics, Key Laboratory of Optoelectronic Devices and Systems of Ministry of Education and Guangdong Province, College of Physics and Optoelectronic Engineering, Shenzhen University, Shenzhen 518060, China

**Keywords:** carbon dot, Fe^3+^, ascorbic acid, cell imaging, molecular detection

## Abstract

High-quality fluorescent probes based on carbon dots (CDs) have promising applications in many fields owing to their good stability, low toxicity, high quantum yield, and low raw material price. The fluorine- and nitrogen-doped fluorescent CDs (NFCDs) with blue fluorescence was successfully synthesized using 3-aminophenol and 2,4-difluorobenzoic acid as the raw material by the hydrothermal method. The NFCDs as probe can be used to directly and indirectly detect Fe^3+^ (detection range: 0.1–150 μM and detection limit: 0.14 μM) and ascorbic acid (AA) (detection range: 10–80 μM and detection limit: 0.11 μM). The NFCDs-based probe shows exceptional selectivity and strong anti-interference for Fe^3+^ and ascorbic acid (AA). In addition, we examined the response of NFCDs to Fe^3+^ and AA in living cells, which showed that the timely use of AA can reduce the effects of iron poisoning. This has important biological significance. This means that using NFCDs as fluorescent probes is beneficial for Fe^3+^ and AA detection and observing their dynamic changes in living cells. Thus, this work may contribute to the study of Fe^3+^- and AA-related diseases.

## 1. Introduction

Fe is an essential trace element in the human body [1], but an overdose of this element may lead to iron poisoning. Importantly, once iron is absorbed by the human body, it can only be excreted in a few ways, except for Fe lost during blood loss. Excess iron in the body is stored in the form of ferritin, which can then be deposited in the liver, heart, and endocrine organs. Furthermore, Fe can cause various diseases, such as cancer, tissue necrosis, Parkinson’s disease, and myocardial infarction disease [2,3,4,5]. Researchers have found that ascorbic acid (AA), also known as vitamin C, can reduce Fe^3+^ to Fe^2+^, reducing the acute diseases caused by excess Fe^3+^. AA has been widely used in health products owing to various benefits such as promoting the formation of antibodies, absorption of Fe, maintaining the activity of sulfhydrylase, and preventing cancer [6,7,8,9]. Therefore, the quantitative detection of Fe^3+^ and AA is of great biological significance.

In the past few decades, there have been a variety of methods used for the quantitative analysis and detection of intracellular and extracellular Fe^3+^, such as atomic absorption spectrometry, electrochemical methods, and inductively coupled plasma in vivo mass spectrometry [10,11,12]. However, these techniques have various disadvantages such as high cost, complicated procedures, narrow detection range, and easy interference from other metal ions during detection. There are also many types of detection methods for AA, such as titrimetry [13], chemiluminescence [14], and capillary electrophoresis [15]. However, these methods also have various limitations such as damage to the sample, high cost, and slow speed. In this context, fluorescence spectroscopy sensing technology based on fluorescent probes have received widespread attention owing to its advantages such as simple operation, low cost, fast response, and high selectivity. In the past few years, many fluorescent probes have been reported that can simultaneously detect Fe^3+^ and AA, including nanocomposites, carbon dots (CDs), and gold nanoclusters. For example, Yang et al. designed a red-emitting gold nanocluster with a precise molecular formula of Au_7_(DHLA)_2_Cl_2_. This rigid structure of probe was destroyed in the presence of Fe^2+^, so it can be used to detect Fe^2+^. The detection range and detection limit of Fe^2+^ were in the range 10–100 μM and 0.2 μM [16]. Although the probe has great advantages in stability and quantum yield, the detection ability of Fe^2+^ needs to be improved. Tai et al. prepared A cysteamine (CA) functionalized copper nanoclusters, which exhibited a detection range of 0–1000 μM and 0–10,000 μM for Fe^3+^ and I^−^, respectively, while the detection limit of Fe^3+^ and I^−^ was 0.92 μM and 3.01 μM, respectively [17]. However, considering the high detection limit for Fe^3+^ and significant cost of precious metals, these techniques are not suitable for mass production. Ungor et al. synthesized an adenosine monophosphate (AMP)-stabilized fluorescent gold nanoclusters for the sensitive and selective detection of Fe^3+^. The detection range and limit of Fe^3+^ reached 10–100 μM and 2.0 μM, respectively [18]. Despite the low detection range, the detection limit still needs to be improved. Shojaeifard et al. designed novel fluorometric penicillamine-capped bimetallic gold-copper nanoclusters. This probe was significantly quenched upon the addition of Fe^3+^ due to the inner filter effect, so it can be used to detect Fe^3+^. The detection range and limit of Fe^3+^ reached 0.5–100 μM and 0.1 μM, respectively. Despite the low detection limit, the detection range still needs to be improved. Shabbir et al. designed a novel carbon quantum dot. This probe was quenched upon the addition of Fe^3+^, so it can be used to detect Fe^3+^. The detection range and limit of Fe^3+^ reached 0–60 ppM and 0.49 ppM, respectively. Although the low detection limit is very excellent, the detection range still needs to be improved. Therefore, research on high-quality fluorescent probes with broad detection ranges for Fe^3+^ and AA is still of great significance. Xu et al. prepared nitrogen-doped graphene quantum dots, which exhibited a detection range of 0.5–50 μM and 6–60 μM for Fe^3+^ and AA, respectively [19]. However, considering the narrow detection range for Fe^3+^ and significant cost of precious metals, these techniques are not suitable for mass production. In addition, although semiconductor quantum dots and nanoclusters possess a greater fluorescence quantum yield and stability, they have a higher toxicity and contain fewer types of surface ligands, which limits the possibility for dynamic imaging of living cells and detection of a wide range of ions and molecules. In comparison, CDs have better chemical and optical properties, such as a high fluorescence quantum yield, good stability, excellent biocompatibility, and strong resistance to bleaching, and are thus widely used for biological imaging and sensing, ion- and small-molecule detection, and other fields [20,21,22]. Therefore, using CDs as fluorescent probes is more beneficial for Fe^3+^ and AA detection and observing their dynamic changes in living cells.

We hope to design a fluorescent carbon dot with a better detection limit and detection range for Fe^3+^ and AA to detect Fe^3+^ and AA in cells. In this work, we used 3-aminophenol and 2,4-difluorobenzoic acid as precursors to synthesize fluorine- and nitrogen-doped fluorescent CDs (NFCDs) with an emission peak wavelength of 428 nm via the hydrothermal method. The NFCDs could directly and indirectly detect Fe^3+^ and AA and exhibited a wider detection range for each. An experiment was subsequently performed to determine the response of NFCDs for Fe^3+^ and AA in living cells, which showed that the timely administration of moderate AA can reduce the effects of iron poisoning, which has important biological implications.

## 2. Experiment Section

### 2.1. Materials

Metal salts (AgNO_3_, NaCl, KI, MnCl_2_·4H_2_O, CaCl_2_, MgCl_2_·6H_2_O, Ba(CH_3_COO)_2_, Pb(CH_3_COO)_2_, ZnCl_2_, FeCl_3_·6H_2_O, CrCl_3_·6H_2_O, CuCl_2_, NaOH, Na_2_HPO_4_, Na_2_SO_3_, Na_2_CO_3_, and NaF), AA, L-Aspartic Acid (ASP), L-Lysine (Lys), Glycine (Gly), L-Glutathione (GSH), L-Histidine (His) and HCl were purchased from Macklin (Shanghai, China). 3-aminophenol was supplied by Macklin (Shanghai, China). 2,4-diflouorbenzoylacetonitrile was purchased from Shanghai Sinopharm Chemical Reagent Co., Ltd. (Shanghai, China).

### 2.2. Instruments

Transmission electron microscopy (TEM, FEI TECNAI G2 F20, Hillsboro, OR, USA), X-ray photoelectron spectroscopy (XPS, Thermo Fisher ESCALAB 250Xi, Waltham, MA, USA), Fourier-transform infrared spectroscopy (FT-IR, Nicolet 5700 spectrometer, Waltham, MA, USA), ultraviolet-visible spectroscopy (UV-Vis, UV-2550 Shimadzu, Osaka, Japan), photoluminescence spectroscopy (PL, Varian Cary Eclipse Agilent, Santa Clara, CA, USA), and fluorescence lifetime measurements (FL, FLS1000 photoluminescence spectrometer) were used to determine the morphology, chemical composition, chemical structure, and optical properties of each sample. A laser-scanning confocal fluorescence microscope (Nikon A1R MP+, Tokyo, Japan and Leica SP8, Wetzlar, Germany) was used for cell imaging.

### 2.3. Synthesis of NFCDs and Measurement of Fluorescence Quantum Yield of NFCDs

During the synthesis of the NFCDs, 0.2 g 3-aminophenol and 0.075 g 2,4-difluorobenzoic acid were added as solutes into a polytetrafluoroethylene-lined autoclave containing 15 mL of water and heated at 180 °C for 12 h.

After the reaction, the reaction container was naturally cooled to room temperature. The solution was removed and placed in a centrifuge tube, centrifuged at 8000 rpm for 5 min, and the supernatant collected. Centrifugation was performed twice as to remove large particles. Then, the sample was dialyzed with a 500 Da membrane and deionized water for three days. The deionized water was changed every 12 h during the dialysis process. The dialysate was filtered through a 0.22 μM polyethersulfone aqueous membrane and lyophilized to obtain the NFCD powder. Then, the absolute quantum yield of NFCDs was measured by photoluminescence spectroscopy.

### 2.4. Quantitative Detection of Fe^3+^ and AA

The same concentration of NFCDs (0.1 mg mL^−1^) was mixed with different known concentrations of Fe^3+^ (0.1–150 μM), and a fluorescence spectrometer was used to measure the emission intensity at 428 nm under excitation at 355 nm. However, to detect AA, NFCDs were added to 150 µM Fe^3+^ and different concentrations of AA (10–80 µM) for 20 min, and the changes in the fluorescence intensity were recorded. The detection limit of the NFCDs for Fe^3+^ and AA were subsequently calculated according to the formula proposed by the International Union of Pure and Applied Chemistry (IUPAC) [23]:L = 3σ/k
where k represents the slope of the graph of fluorescence intensity versus Fe^3+^ or AA concentration, and σ represents the standard deviation of the background sample. The standard deviation was measured by measuring the emission wavelength of the blank sample multiple times at a fluorescence wavelength of 428 nm.

### 2.5. Cell Imaging and Cell Viability

Human esophageal cancer cells (KYSE-150 cells) were placed in a 96-well plate and incubated for one day, and then a mixture of different concentrations of the CDs and culture medium was added to the 96-well plate for another 12 h. The relative viability of the KYSE-150 cells was determined using the CCK-8 assay. Finally, the absorbance of the lysed cells was recorded at 570 nm using a microplate reader. To reduce the error, each experiment had six comparison data.

Fluorescence quenching was recorded after exogenous addition of 100 μL Fe^3+^ (concentration: 10 mM) to the KYSE-150 cells stained with NFCDs for 20 min. After the cell fluorescence was stable, 200 μL of AA (concentration: 10 mM) was added and the fluorescence recovery recorded.

## 3. Results and Discussion

### 3.1. Characterization of Nanoprobe NFCDs

In this study, blue emission NFCDs were synthesized by the hydrothermal method at 180 °C for 12 h (Figure 1). To further investigate the optimal synthesis conditions, we used different ratios of 3-aminophenol and 2,4-difluorobenzoic acid to synthesize different fluorescent carbon dots (Figure 1a). It is obvious that the fluorescent carbon dots have the best fluorescence when the ratio of 3-aminophenol and 2,4-difluorobenzoic acid is 8/3 strength. Therefore, NFCDs are synthesized with this ratio. And its quantum yield is 12.66% (Figure 1b). The TEM results revealed the morphology and size distribution of the NFCDs (Figure 1c). The NFCDs consist of approximately 2.60 nm sized nanoparticles (Figure 1c,d).

In addition, FT-IR spectroscopy revealed the surface functional groups of the NFCDs (Figure 2a). The characteristic absorption peaks at 3448 and 3228 cm^−1^ correspond to the typical stretching vibrations of -COH and -NH_2_, respectively, which presumably originate from 3-aminophenol after the reaction. The characteristic absorption band at 1630 cm^−1^ was attributed to the strong stretching vibration of asymmetrical C=O [24]. The peaks at 1410 cm^−1^ were attributed to the typical stretching vibrations of C=N. In addition, the two absorption peaks at 990 and 1260 cm^−1^ correspond to the C-F stretching vibrations [24,25]. To further elucidate the chemical structure, XPS was used to analyze the elemental contents and chemical bonds. As shown in Figure 2b, the NFCDs are composed of four elements: C, O, N, and F. The corresponding binding energies at 285.08 eV, 531.08 eV, 400.08 eV, and 600.08 eV originate from C 1s, O 1s, N 1s, and F 1s [26]. The contents of these elements are 74.22%, 14.33%, 10.11%, and 1.35%, respectively. The O 1s XPS spectrum shows the presence of C=O (531.2 eV) and C-O (531.8 eV), as shown in Figure 2c [26]. Figure 2d shows the high-resolution C 1s spectrum, which exhibits three peaks: C=O (286.3 eV), C-O (285.6 eV), and C-C/C-N (284.8 eV) [24,25]. The N 1s XPS spectrum is displayed in Figure 2e, which includes three peaks at 399.8 eV, and 401.7 eV, corresponding to pyridinic N, C-N=C, and Amino N (Figure 2e), respectively [27]. The F 1s XPS spectrum shows two peaks at 686.48 eV and 687.28 eV, which corresponds to semi-ionic C-F and covalent C-F (Figure 2f) [24]. Therefore, the XPS and FT-IR analyses confirmed that F and N were doped into NFCDs.

The optical characteristics of NFCDs were investigated by measuring the absorption, excitation and emission spectra, which are collected in Figure 3. The peaks at 242 nm and 308 nm are attributed to the π-π* transition of the conjugated sp^2^ domains from the C core. The peak at 427 nm corresponded to the n-π* transition. As shown in Figure 3a, the NFCDs display a blue emission peak at 428 nm. In Figure 3b, the 3D spectrum shows the fluorescence emission spectrum of NFCDs is dependent in different excitation spectrum. The fluorescence spectrum of NFCD is dependent of excitation wavelength.

### 3.2. Selectivity toward Fe^3+^ and AA

To use NFCDs as sensors to detect the presence of Fe^3+^ in the environment, we examined the stability of the salt solution, photostability, pH stability of NFCDs, and reaction sensitivity. As shown in Figure 4a, the fluorescence intensity remains stable in a NaCl solution at a concentration of 0–200 µM, which suggests a good stability in a salt solution. Under continuous irradiation with a 365 nm light source, the fluorescence intensity remained relatively consistent for approximately 1 h, which confirmed the optical stability of the NFCDs (Figure 4b). As shown in Figure 4c, NFCDs are sensitive to both strong alkalis and acids, although the fluorescence intensity remained relatively stable within the range of pH 3–12. Although the performance decreased under strong acidic and basic conditions, most solutions tested in practice are weakly basic or acidic, and thus the NFCDs are very suitable as Fe^3+^ sensors. The fluorescence intensity of NFCDs almost remained unchanged after 10 min reaction with Fe^3+^, which shows the NFCDs processed high sensitivity to Fe^3+^ (Figure 2d).

The selectivity of NFCDs against different ions was examined by measuring the fluorescence response to ions. Compared to other ions, the NFCDs exhibited a remarkable fluorescence quenching for Fe^3+^. Figure 5a shows that Cd^2+^, Zn^2+^, Pb^2+^, Cu^2+^, Ag^+^, Mn^2+^, Ca^2+^, Mg^2+^, HCO_3_^−^, CO_3_^2−^, HPO_4_^2−^, Na^+^, AA, LAC, LLY, Gly, GSH and LHI caused almost no response, which demonstrates the exceptional selectivity of NFCDs for Fe^3+^ over other ions and amino acids. Figure 5b shows the intensity after adding different ions and amino acids in the solution of Fe^3+^ co-existing with NFCDs. Notably, only AA added to the reaction system restored the fluorescence intensity. The results verified that the NFCDs have a very strong anti-interference capability and can indirectly detect AA.

In addition, the absorption spectrum and fluorescence lifetime were used to analyze the quenching mechanism of the NFCDs (Appendix A, Figure 5c,d). Common causes of decreased fluorescence usually include photo-induced electron transfer, fluorescence resonance energy transfer, static quenching effect (SQE), and dynamic quenching effect [28,29,30,31]. Since the fluorescence lifetime has an important relationship with the fluorescence quenching mechanism, we tested the fluorescence lifetime of NFCDs in different concentrations of Fe^3+^. Since the fluorescence lifetime of the probe remains unchanged, the quenching mechanism of NFCDs is considered to be static quenching. In addition, the absorption spectra of Fe^3+^, NFCDs and Fe^3+^/NFCDs were also measured (Appendix A). It can be seen that the absorption peak of NFCDs absorption spectrum shifted after Fe^3+^ was added, which means that new substances were produced. This is because NFCDs combined with Fe^3+^ to form a non-luminous ground state complex. This further confirmed that the fluorescence quenching mechanism of the NFCDs was SQE. In addition, when AA was introduced into the reaction, the fluorescence was restored and τ3, τ4 and τAvg2 nearly remained unchanged, which further confirmed that SQE is the cause of fluorescence reduction (Appendix A). As this quenching is reversible, an experiment could be designed to indirectly detect AA using this characteristic. There was no electron transfer between the NFCDs and Fe^3+^; therefore, when AA was added to this reaction, Fe^3+^ was reduced by AA to Fe^2+^, which released the NFCDs and restored the fluorescence. After 300 μM AA were added to the reaction system of 300 μM Fe^3^+ and NFCDs, we examined the relationship between the corresponding fluorescence intensity and reaction time (the wavelength was 428 nm) (Appendix A). The fluorescence intensity stabilized and was restored after 20 min. Thus, the experimental results showed that the NFCDs have very good direct or indirect selectivity toward Fe^3+^ and AA.

### 3.3. Detection Range and Limit of Fe^3+^ and AA for the NFCDs-Based Probe

Previous experiments have shown that NFCDs have excellent selectivity for Fe^3+^ and AA. Therefore, the relationship between the fluorescence intensity of NFCDs and the change in the Fe^3+^ and AA concentration was further studied. In order to exclude the influence of the inner filter effect on the NFCDs, the fluorescence intensity of NFCDs was corrected by the following formula:$$F_{C}=F_{M}\times 10\times \left(A_{EX}+A_{EM}\right)/2$$
where $\;F_{C}\;$ represents the corrected intensity at 428 nm and $F_{M}$ represents the registered intensity at 428 nm. Further, $A_{EX}$ and $A_{EM}$ represent the absorbance of Fe^3+^ at 355 nm and 428 nm, respectively. The emission spectrum distribution of 0.1 mg mL^−1^ NFCDs after quenching with different Fe^3+^ concentrations is shown in Figure 6a. Figure 6b shows the linear relationship between the Fe^3+^ concentration and (F_0_ − F)/F_0_ (where F_0_ represents the fluorescence intensity of NFCDs without Fe^3+^ and F represents the corrected fluorescence intensity of NFCDs with different concentrations of Fe^3+^). The linear relationship equation was determined as:$$Y_{1}=0.02065+0.00242\;X_{1}$$
where $X_{1}\;$ represents the concentration of Fe^3+^ (μM) and $Y_{1}$ represents (F_0_ − F)/F_0_. The R^2^ was found to be 0.99578, indicating a very good linear fit. The detection limit was approximately 0.14 μM and the detection range was between 0.2–150 μM.

In addition, because of the presence of AA, Fe^3+^ can be reduced to Fe^2+^. Therefore, the fluorescence intensity changes were recorded after adding NFCDs to Fe^3+^ (150 μM) and different concentrations of AA (10–80 μM) for 20 min (Figure 6c). Figure 6d shows the linear relationship between the corrected fluorescence intensity of the emission spectrum at 428 nm and the concentration of AA, from which the following equation was obtained:$$Y_{2}=0.37176-0.00311\;X_{2}$$
where $X_{2}$ represents the concentration of AA and $Y_{2}$ represents (F_1_ − F_2_)/F_1_ (where F_1_ represents the corrected fluorescence intensity of the NFCDs with 10 μM AA and F_2_ represents the corrected fluorescence intensity with different concentrations of AA). The detection limit and detection range were 0.11 μM and 10–80 μM, respectively. Compared with other types of fluorescent probes (Table 1), which have a narrow detection range, NFCDs are excellent for the selective detection of AA and Fe^3+^. In general, NFCDs have very good potential for the quantitative detection of Fe^3+^ and AA.

### 3.4. Cell Imaging of NFCDs in Response to Fe^3+^ and AA

As shown in Appendix A, the cells with the NFCDs at a concentration of 120 μg mL^−1^ maintained very good activity, indicating that NFCDs have low cytotoxicity and are suitable for cell imaging. Therefore, cells were stained with 300 μg mL^−1^ NFCDs for 20 min to measure the NFCD response toward intracellular Fe^3+^ and AA. To show the overlapping cell imaging information from the bright and fluorescence field, (Figure 7c,f,i) represent the mixed blue fluorescence and bright field imaging. The cells without any added substance, those with 100 μL Fe^3+^ (concentration: 10 mM), and those with 200 μL AA (concentration: 10 mM) added after reacting for 5 min are shown in Figure 7a–i, respectively. The cells exhibit an obvious darkening and brightening process. These results indicate that AA can quickly enter the cell and react with Fe^3+^ to release the NFCDs.

## 4. Conclusions

In this work, we synthesized a new type of fluorescent NFCD by a simple hydrothermal method using 3-aminophenol and 2, 4-difluorobenzoic acid. Due to the doping of N and F elements, the NFCDs had blue fluorescence (428 nm), showed excellent stability [37], exhibited a high selectivity for Fe^3+^ and AA, and have a broad detection range. The detection range of Fe^3+^ and AA have reached 0.2–150 μM and 10–80 μM. The detection limit of Fe^3+^ and AA have reached 0.14 uM and 0.11 μM. In addition, experiments were conducted on the response of NFCDs to Fe^3+^ and AA in living cells. The results showed that when too much Fe^3+^ was accidentally consumed, an appropriate amount of AA over time will slow down the effects of iron poisoning, which illustrates the biological significance of this study. NFCDs are of great significance for the detection of Fe^3+^ and AA in living cells. However, due to the limitation of high detection limit and complex cell environment, NFCDs still cannot detect spontaneous Fe^3+^ in alive cells. It also limits wide application of NFCDs in the biomedical field.

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
