# Peer review of "Fluorine-Nitrogen-Codoped Carbon Dots as Fluorescent Switch Probes for Selective Fe(III) and Ascorbic Acid Sensing in Living Cells"

_molecules, 2022, doi:10.3390/molecules27196158_

Round 1

Reviewer 1 Report (Previous Reviewer 1)

The authors re-uploaded their work on detecting Fe3+ ions and ascorbic acid via fluorescence quenching of carbon dots.

Most errors have been corrected, but I still disagree with the authors on two parts:

1) The correction of the fluorescence spectra is necessary. The authors present some example UV-Vis spectra, which include the optical signal of the 10 uM Fe3+ BUT the whole range is 0.1-150 uM and towards the end of the range the selected transition metal ion show high absorbance! For this reason, correction is needed!!!!

2) Moreover, in chapter 3.3 (LOD calculation), the detection limit is 0.14 uM for the Fe3+, in this case, the dynamic range cannot start at 0.1 uM because it is under the detectable amount!!!!!

Without these corrections, I do not support the publication at all because the manuscript contains serious faults!

Author Response

Reviewer #1:

The authors re-uploaded their work on detecting Fe3+ ions and ascorbic acid via fluorescence quenching of carbon dots.

Most errors have been corrected, but I still disagree with the authors on two parts:

1 The correction of the fluorescence spectra is necessary. The authors present some example UV-Vis spectra, which include the optical signal of the 10 uM Fe3+ BUT the whole range is 0.1-150 uM and towards the end of the range the selected transition metal ion show high absorbance! For this reason, correction is needed!!!!

Response 1:

Thank you for your valuable advice. We have changed some text and picture. The revised parts were showed as follows:

In order to exclude the influence of the inner filter effect on the NFCDs, the fluorescence intensity of NFCDs was corrected by the following formula:

whererepresents the corrected intensity at 428nm and  represents the registered intensity at 428nm. Besides,  and  represent the absorbance of Fe3+ at 355nm and 428nm, respectively.

Fig. 6b shows the linear relationship between the Fe3+ concentration and (F0-F)/F0 (where F0 represents the fluorescence intensity of NFCDs without Fe3+ and F represents the corrected fluorescence intensity of NFCDs with different concentrations of Fe3+). The linear relationship equation was determined as:

where represents the concentration of Fe3+ (μM) and  represents (F0-F)/F0. The R2 was found to be 0.99578, indicating a very good linear fit. The detection limit was approximately 0.14 μM and the detection range was between 0.2–150 μM.

Fig. 6d shows the linear relationship between the corrected fluorescence intensity of the emission spectrum at 428 nm and the concentration of AA, from which the following equation was obtained:

where  represents the concentration of AA and  represents (F1-F2)/F1 (where F1 represents the corrected fluorescence intensity of the NFCDs with 10 μM AA and F2 represents the corrected fluorescence intensity with different concentrations of AA). The detection limit and detection range were 0.11 μM and 10–80 μM, respectively.

Fig. 6 (a) The normalized fluorescence spectra of NFCDs with different Fe3+ concentrations. (b) Linear relationship between the corrected fluorescence intensity and Fe3+ concentration (F0 is the fluorescence intensity without Fe3+ and F is the corrected fluorescence intensity at different Fe3+ concentrations). (c) Fluorescence spectra after adding AA to the solution of Fe3+ (150 μM) and NFCDs for 20 min. (d) Linear relationship between the corrected fluorescence intensity and AA concentration (F1 is the corrected fluorescence intensity of NFCDs/Fe3+ without AA and F2 is the corrected fluorescence intensity at different AA concentrations).

2 Moreover, in chapter 3.3 (LOD calculation), the detection limit is 0.14 uM for the Fe3+, in this case, the dynamic range cannot start at 0.1 uM because it is under the detectable amount!

Response 2:

Thank you for your valuable suggestion. We re-measured the fluorescence intensity at 0.2 μM, and modified some texts and pictures. The modified parts are as follows:

The detection limit was approximately 0.14 μM and the detection range was between 0.2–150 μM.

The detection range of Fe3+ and AA have reached 0.2-150 μM and 10-80 μM.

Without these corrections, I do not support the publication at all because the manuscript contains serious faults!

Reviewer 2 Report (Previous Reviewer 2)

The reviewer’s comments are superficially addressed without performing a careful revision. The following concerns should be addressed

Major comments

Comment 1: In line 161 what does it mean by “the ratio of 3 and 4”

Comment 2: Regarding the quantum yield (QY) calculation, the authors just provided the value in response to my comment. The authors should provide the method of QY calculation in experimental section and should provide the QY value is results.

Comment 3: In XPS analysis, I have suggested to correct the peak assignment for both C 1s and N 1s. However, the N 1s peak assignment is still wrong. There are only two peaks in figure 2e, but the authors described 3 peaks in the text.

Comment 4: Regarding Table S3, the authors should place it in the main manuscript to show the potential of the current probe. There is no need to explain it in detail.  

Comment 5: Regarding my comment on providing the possible reason for the better performance of the probe, the authors just provided a response to my comment. However, the authors did not provide any explanation in the manuscript. I suggest the authors to provide some explanation in the main manuscript.  

Minor comments

To my surprise the authors addressed only one comment and deleted the remaining without addressing. As these are minor comments, I am providing them again.

1.      In line 145, the reaction time has given as 24 h whereas it is presented as 12 h in other places

2.      The title of the section 3.1 is “Design strategies and characterization of nanoprobe NFCDs” However, there is no design strategy in the following explanation

3.      In figure 2 caption, it should be FTIR “spectrum” not spectra

4.      In FTIR explanation, there no mention of C-F peaks that are shown in Figure

5.      The usage of the acronym NFCDs should be consistent throughout the manuscript. In many cases the authors used CDs instead of NFCDs which is confusing.

6.      In case of the data in Fig 5b, how much concentration of iron and other substances were used ? authors should mention this

7.      In figure 6a caption, it should be spectra not spectrum

8.      There are several typographical and grammatical errors. The manuscript should be thoroughly revised for language errors.

Author Response

Reviewer #2:

The reviewer’s comments are superficially addressed without performing a careful revision. The following concerns should be addressed

Major comments

Comment 1: In line 161 what does it mean by “the ratio of 3 and 4”

Response 1:

Thank you for your valuable advice. We have changed some text and picture. The revised parts were showed as follows:

It is obvious that the fluorescent carbon dots have the best fluorescence when the ratio of 3-aminophenol and 2, 4-difluorobenzoic acid is 8/3 strength.

Comment 2: Regarding the quantum yield (QY) calculation, the authors just provided the value in response to my comment. The authors should provide the method of QY calculation in experimental section and should provide the QY value is results.

Response 2:

Thank you for your valuable advice. We have changed some text and picture. The revised parts were showed as follows:

2.3. Synthesis of NFCDs and Measurement of Fluorescence QuantumYield of NFCDs

Then, the absolute quantum yield of NFCDs was measured by photoluminescence spectroscopy.

And its quantum yield is 12.66% (Fig. 1b). The TEM results revealed the morphology and size distribution of the NFCDs (Fig. 1c). The NFCDs consist of approximately 2.60 nm sized nanoparticles (Fig. 1c, d).

Figure 1. (a) Fluorescence intensity spectra of 3-aminophenol and 2,4-difluorobenzoic acid reacted at different ratios. (b) Fluorescence quantum yield of NFCDs when the ratio of -aminophenol and 2,4-difluorobenzoic acid is 8/3. (c) TEM image of NFCDs. (d) Particle size distribution diagram of NFCDs.

Comment 3: In XPS analysis, I have suggested to correct the peak assignment for both C 1s and N 1s. However, the N 1s peak assignment is still wrong. There are only two peaks in figure 2e, but the authors described 3 peaks in the text.

Response 3:

Thank you for your valuable advice. We have changed some text and picture. The revised parts were showed as follows:

Fig. 2d shows the high-resolution C 1s spectrum, which exhibits three peaks: C=O (286.3 eV), C-O (285.6 eV), and C-C/C-N (284.8 eV).[24, 25] The N 1s XPS spectrum is displayed in Fig. 2e, which includes three peaks at 399.8 eV, and 401.7 eV, corresponding to pyridinic N, C-N=C, and Amino N (Fig. 2e), respectively

Comment 4: Regarding Table S3, the authors should place it in the main manuscript to show the potential of the current probe. There is no need to explain it in detail. 

Response 4:

Thanks for your suggestion, Table. S3 has been put into the main text. The revised parts were showed as follows:

Table. 1 Comparison of several fluorescent probes for Fe3+ and AA described in recent reports.

Types of fluorescent probes

Detection range of Fe3+(μM)

Detection range of AA(μM)

Reference

Carbon dot

0-0.38

0-0.78

[32]

Metal-organic framework

5-60

1-20

[33]

Carbon dot

0-145

0–150

[34]

nanoclusters

0.5-80

0.2-80

[35]

Carbon dot

0-100

0-100

[36]

Carbon dot

0.75-125

0.25-30

[37]

Carbon dot

0-150

10-80

This work

Reference:

  1. 32. Raveendran, V.; Suresh Babu, A. R.; Renuka, N. K., Mint leaf derived carbon dots for dual analyte detection of Fe(iii) and ascorbic acid. RSC Adv. 2019, 9, (21), 12070-12077.
  2. Wang, H.; Wang, X.; Kong, R.-M.; Xia, L.; Qu, F., Metal-organic framework as a multi-component sensor for detection of Fe3+, ascorbic acid and acid phosphatase. Chin. Chem. Lett. 2021, 32, (1), 198-202.
  3. Lv, X.; Man, H.; Dong, L.; Huang, J.; Wang, X., Preparation of highly crystalline nitrogen-doped carbon dots and their application in sequential fluorescent detection of Fe3+ and ascorbic acid. Food Chem. 2020, 326, 126935.
  4. Yang, X.; Yang, J.; Zhang, M.; Wang, Y.; Zhang, B.; Mei, X., Tiopronin protected gold-silver bimetallic nanoclusters for sequential detection of Fe3+ and ascorbic acid in serum. Microchem. J. 2022, 174, 107048.
  5. Bandi, R.; Devulapalli, N. P.; Dadigala, R.; Gangapuram, B. R.; Guttena, V., Facile Conversion of Toxic Cigarette Butts to N,S-Codoped Carbon Dots and Their Application in Fluorescent Film, Security Ink, Bioimaging, Sensing and Logic Gate Operation. ACS Omega 2018, 3, (10), 13454-13466.
  6. Zhang, S.; Zhang, C.; Shao, X.; Guan, R.; Hu, Y.; Zhang, K.; Liu, W.; Hong, M.; Yue, Q., Dual-emission ratio fluorescence for selective and sensitive detection of ferric ions and ascorbic acid based on one-pot synthesis of glutathione protected gold nanoclusters. RSC Adv. 2021, 11, (28), 17283-17290.

Comment 5: Regarding my comment on providing the possible reason for the better performance of the probe, the authors just provided a response to my comment. However, the authors did not provide any explanation in the manuscript. I suggest the authors to provide some explanation in the main manuscript. 

Response 5:

Thank you for your valuable advice. We have changed some text and picture. The revised parts were showed as follows:

Due to the doping of N and F elements, the NFCDs had blue fluorescence (428 nm), showed excellent stability [38], exhibited a high selectivity for Fe3+ and AA, and have a broad detection range

Minor comments

To my surprise the authors addressed only one comment and deleted the remaining without addressing. As these are minor comments, I am providing them again.

  1. In line 145, the reaction time has given as 24 h whereas it is presented as 12 h in other places

Response 1:

Thank you for your valuable advice. We have changed some text. The revised parts were showed as follows:

In this study, blue emission NFCDs were synthesized by the hydrothermal method at 180 °C for 12 h (Scheme 1).

  1. The title of the section 3.1 is “Design strategies and characterization of nanoprobe NFCDs” However, there is no design strategy in the following explanation

Response 2:

Thank you for your valuable suggestion. We have changed some text. The revised parts were showed as follows:

3.1. Characterization of nanoprobe NFCDs

  1. In figure 2 caption, it should be FTIR “spectrum” not spectra

Response 3:

Thank you for your valuable advice. We have changed text. The revised parts were showed as follows:

Figure 2. (a) FT-IR spectrum and (b) XPS survey spectrum of blue emissive CDs, and the corresponding high-resolution (c) O 1s, (d) C 1s, (e) N 1s, and (f) F 1s XPS spectra.

  1. In FTIR explanation, there no mention of C-F peaks that are shown in Figure

Response 4:

Thank you for your valuable advice. We have changed some text and picture. The revised parts were showed as follows:

In addition, the two absorption peaks at 990 and 1260 cm−1 correspond to the C-F stretching vibrations.

  1. The usage of the acronym NFCDs should be consistent throughout the manuscript. In many cases the authors used CDs instead of NFCDs which is confusing.

Response 5:

Thank you for your valuable advice. We have changed some text. The revised parts were showed as follows:

Figure 3. (a) Emission spectrum (blue line) under 355 nm excitation, excitation spectrum (red line) under 428 nm emission, and the UV-vis absorption (black dotted line) spectrum of NFCDs. Inset: photograph of the aqueous solution of NFCDs acquired under 365 nm light irradiation.

The UV-vis absorption spectrum of the NFCDs exhibited three characteristic absorption peaks at 242, 308, and 427 nm.

In addition, the absorption spectra of Fe3+, NFCDs and Fe3+/NFCDs were also measured (Fig.S1). It can be seen that the absorption peak of NFCDs absorption spectrum shifted after Fe3+ was added, which means that new substances were produced. This is because NFCDs combined with Fe3+ to form a non-luminous ground state complex

where  represents the concentration of AA and  represents (F1-F2)/F1 (where F1 represents the fluorescence intensity of the NFCDs with 10 μM AA and F2 represents the fluorescence intensity with different concentrations of AA). The detection limit and detection range were 0.11 μM and 10–80 μM, respectively. Compared with other types of fluorescent probes (Table 1), which have a narrow detection range, NFCDs are excellent for the selective detection of AA and Fe3+. In general, NFCDs have very good potential for the quantitative detection of Fe3+ and AA.

These results indicate that AA can quickly enter the cell and react with Fe3+ to release the NFCDs.

  1. In case of the data in Fig 5b, how much concentration of iron and other substances were used? authors should mention this

Response 6:

Thank you for your valuable advice. We have changed text. The revised parts were showed as follows:

(b) Fluorescence intensity of NFCDs after adding different ions and amino acids into the reaction system containing Fe3+ and NFCDs (concentration: 1 mM).

  1. In figure 6a caption, it should be spectra not spectrum

Response 7:

Thank you for your valuable advice. We have changed the text. The revised parts were showed as follows:

Fig. 6 (a) The normalized fluorescence spectra of NFCDs with different Fe3+ concentrations.

  1. There are several typographical and grammatical errors. The manuscript should be thoroughly revised for language errors.

Response 8:

Thank you for your valuable advice. We have changed some text. The revised parts were showed as follows:

the NFCDs had blue fluorescence (428 nm)

The results showed that when too much Fe3+ was accidentally consumed, an appropriate amount of AA over time will slow down the effects of iron poisoning, which illustrates the biological significance of this study.

Round 2

Reviewer 1 Report (Previous Reviewer 1)

The recommended modifications are done. I support the publication in the present form.

This manuscript is a resubmission of an earlier submission. The following is a list of the peer review reports and author responses from that submission.

Round 1

Reviewer 1 Report

The presented work is after a previous review process. The theme of the article is good but I still found some false interpretations.

1) Related to fluorescence quenching. The authors showed that the fluorescence quenching is totally static based on the lifetime measurements. During the static quenching, the KSV (Stern-Volmer constant) can be considered as an association constant (Ka). In this case, the calculation of the collision quenching constant (lines 230-243) is a colossal mistake and unnecessary!

2) The unsatisfactory interpretation of the relationship between the fluorescence and UV-Vis spectra. It is well-known that Fe3+ has a large absorbance in the studied spectral range as can be seen on Fig. S1. In the case of colorful materials, the spectral correction from the inner-filter effect and self-absorption is cannot be avoided!!! The use of the following equation* is recommended:

                        Icorrected = Imeasured × 10(AEX + AEM)/2

 Icorrected = corrected intensity at the selected wavelength

Imeasured = registered PL intensity at the selected wavelength

AEX and AEM = the absorbance of the sample at the excitation and emission wavelength, respectively.

* J. R. Lakowicz: Principles of Fluorescence Spectroscopy

3) Besides, the quality of the Introduction is also poor. The detection of the Fe3+ ions in liquid samples, as well as biological tissues, is a well-known and widely studied work for several years ago. I suggest extending it with some relevant articles, which focus on the sensing of this ion via a quenching mechanism. 

https://doi.org/10.1039/C7TC00724H

https://doi.org/10.1016/j.sbsr.2019.100319

http://dx.doi.org/10.1016/j.colsurfb.2017.04.013

https://doi.org/10.1016/j.saa.2018.10.042

https://doi.org/10.3390/ma14247604

After the suggested major revision of the manuscript, I will support the publication.

Reviewer 2 Report

In the present manuscript entitled “Fluorine-nitrogen-codoped carbon dots as fluorescent switch probes for selective Fe(â…¢) and ascorbic acid sensing in living cells”, the authors described the preparation and characterization of F and N doped carbon dots and applied them in Fe(III) and ascorbic acid detection. There are several papers on application of carbon dots for Fe(III) and ascorbic acid detection. Hence the novelty of the present work lies in the preparation of F and N doped CDs. However, the authors did not focus much on this. Further the manuscript writing is superficial without any depth explanation of their results or concepts. Hence, I feel the manuscript will not add any new knowledge to the field and suggest its rejection.

Major comments

1.      preparation of NFCDs is the novel point of this work. This should be thoroughly studied. Data related to the optimization of synthetic conditions should be provided and more explanation should be provided on this topic in the main text.

2.      Fluorescence Quantum yield should be calculated

3.      In XPS analysis, the peak assignments for C1s and N 1s are not in accordance with the peaks shown in corresponding figure.

4.      I am surprised how the authors calculated the stern volmer constant from the plot of (F0-F)/F vs Fe3+ concentration (Figure S1a). The actual Stern volmer plot is F0/F vs Fe3+ concentration. This questions the integrity of the work.

5.      The absorption spectrum of the same sample (NFCDs) provided in Figure S1b is not matching with that provided in Figure 3a. This questions the reliability of the work.

6.       Comparing the detection performances of other probes in the introduction section looks awkward. This should be done in the results and discussion section. Table S3 should be brought to the main text

7.      What might be the reason for the better performance of the NFCDs probe ? please provide some explanation.

Minor comments

1.      In line 145, the reaction time has given as 24 h whereas it is presented as 12 h in other places

2.      The title of the section 3.1 is “Design strategies and characterization of nanoprobe NFCDs” However, there is no design strategy in the following explanation

3.      In figure 2 caption, it should be FTIR “spectrum” not spectra

4.      In FTIR explanation, there no mention of C-F peaks that are shown in Figure

5.      The usage of the acronym NFCDs should be consistent throughout the manuscript. In many cases the authors used CDs instead of NFCDs which is confusing.

6.      In case of the data in Fig 5b, how much concentration of iron and other substances were used ? authors should mention this

7.      In figure 6a caption, it should be spectra not spectrum

8.      There are several typographical and grammatical errors. The manuscript should be thoroughly revised for language errors.